# Insights into COVID-19: Perspectives on Drug Remedies and Host Cell Responses

**DOI:** 10.3390/biom13101452

**Published:** 2023-09-26

**Authors:** Ahmed M. Awad, Kamryn Hansen, Diana Del Rio, Derek Flores, Reham F. Barghash, Laura Kakkola, Ilkka Julkunen, Kareem Awad

**Affiliations:** 1Department of Chemistry, California State University Channel Islands, Camarillo, CA 93012, USA; 2Institute of Chemical Industries Research, National Research Centre, Dokki, Cairo 12622, Egypt; 3Institute of Biomedicine, Faculty of Medicine, University of Turku, 20014 Turku, Finland; 4Clinical Microbiology, Turku University Hospital, 20521 Turku, Finland; 5Department of Therapeutic Chemistry, Institute of Pharmaceutical and Drug Industries Research, National Research Center, Dokki, Cairo 12622, Egypt

**Keywords:** coronavirus, COVID-19, target cell responses, drug design, nucleoside analogs, ACE2

## Abstract

In light of the COVID-19 global pandemic caused by SARS-CoV-2, ongoing research has centered on minimizing viral spread either by stopping viral entry or inhibiting viral replication. Repurposing antiviral drugs, typically nucleoside analogs, has proven successful at inhibiting virus replication. This review summarizes current information regarding coronavirus classification and characterization and presents the broad clinical consequences of SARS-CoV-2 activation of the angiotensin-converting enzyme 2 (ACE2) receptor expressed in different human cell types. It provides publicly available knowledge on the chemical nature of proposed therapeutics and their target biomolecules to assist in the identification of potentially new drugs for the treatment of SARS-CoV-2 infection.

## 1. Introduction

The emergence of severe acute respiratory syndrome coronavirus 2 (SARS-CoV-2) in Wuhan, China, at the end of 2019 has proven to be capable of causing a severe pneumonia, and the disease is referred to as coronavirus disease 2019 (COVID-19). SARS-CoV-2 is a member of the *coronaviridae* family of viruses belonging to the genus *beta*-coronavirus [1]. On 11 March 2020, the global spread of COVID-19 led to the World Health Organization’s (WHO) escalation of the virus to that of a pandemic. As of January 2023, there has been 680 million confirmed cases, 6.7 million deaths, and more than 13 billion vaccine doses administered [2]. From COVID-19’s inception, researchers have sought to understand the factors that trigger severe illness. Yet, the development of severe symptoms does not appear to be unilaterally attributed to the viral load or the spread of the virus in the host; rather, the severity of the disease may involve host cell responses. As such, researchers are investigating the repurposing of pre-existing antiviral drugs, notably nucleoside analogs (NAs), along with antibiotics, corticosteroids, and other options for SARS-CoV-2 therapy.

### 1.1. Classification of Coronaviruses

The word “corona” in Latin means “crown”. Coronaviruses received this name due to their crown-like surface glycoproteins, commonly known as spike proteins. Seven coronaviruses (HCoVs) cause disease in humans, including the novel SARS-CoV-2, HCoV-229E, HCoV-NL63, HCoV-OC43, HCoV-HKU1, Middle East Respiratory Syndrome (MERS-CoV), and Severe Acute Respiratory Syndrome (SARS-CoV). These human viruses are responsible for respiratory tract infections that present as an array of symptoms ranging in severity from the common cold to pneumonia (Table 1) [3]. Since the beginning of the 21st century, three of the seven HCoVs, have crossed species, making humans susceptible to a deadly pneumonia. Of the seven HCoVs, MERS-CoV, SARS-CoV, and SARS-CoV-2, are considered highly pathogenic with an increased risk of mortality [4]. There are four coronavirus genera: alpha-, beta-, gamma-, and delta-coronaviruses (Table 1). The α- and β-CoVs are found in mammals, while the γ- and δ-CoVs are predominantly found in birds [5,6,7,8].

### 1.2. Structure of Coronaviruses

A coronavirus is a positive-sense RNA virus composed of a lipid bilayer and envelope proteins. Within the SARS-CoV-2 virion, there are four main structural proteins known as the spike glycoprotein (S), which mediates binding of the virus to host ACE2 receptors; the transmembrane protein (M); the envelope membrane protein (E); and the nucleocapsid protein (N) (Figure 1). The nucleocapsid consists of genomic RNA, which contains open reading frames (ORFs) [4,9,10]. Notably, the M protein plays a central role in the assembly and budding via interaction with the S, E, and N proteins [9,10].

### 1.3. Nonstructural Proteins in CoVs

Nonstructural proteins (nsps) are important for the replication of the virus. In contrast, structural and accessory ORF proteins are involved in the assembly of the virus and interference with the host immune responses, respectively [9,10]. Examples of the functions of nsps, including nsp12, an RNA-dependent RNA polymerase (RdRp) that is crucial to RNA synthesis and virus replication, are summarized in Table 2 [3,9,10]. Nsp12 forms a complex with the co-factors nsp7 and nsp8, which act as a primase and catalyze the synthesis of viral RNA [11,12].

## 2. Viral Entry and Host Cell Responses

### 2.1. Infection and Transmission

SARS-CoV-2 can enter the human body either by inhalation or by coming into contact with respiratory droplets containing viral particles, typically from someone sneezing, coughing, or talking [13,14]. Since the start of this pandemic, hundreds of variants have evolved, which has affected the outcome of the pandemic. In spring 2020, SARS-CoV-2 became more infectious and transmissible due to mutations in its spike (S) protein as well as in the RNA-dependent RNA polymerase (RdRp), and these variants gained the capacity to overrun the original SARS-CoV-2 Wuhan strain [15,16]. These variants, including Alpha, Beta, Delta, Kappa and Omicron and their many sublineages, swept across the world. The variants differ in their ability to replicate, transmit, and induce host cell responses in human cells [17,18,19,20]. It was later observed that compared to the early pandemic strains, Omicron variants showed a weaker replication efficiency and a slower interferon response, which indicated that accumulated mutations may contribute to viral adaptation [18,21,22].

### 2.2. ACE2 Receptor and Mediation of Virus Host Cell Entry

To infect humans, SARS-CoV-2, like the earlier SARS-CoV, uses the spike (S) protein to bind to its cell surface receptor, the angiotensin-converting enzyme 2 receptor (ACE2) [12]. After binding to its receptor, the virus is endocytosed, the viral genome is released, and viral RNA is translated into nonstructural proteins; these compose the replication–transcription complex (RTC) [23]. ACE2 is expressed in the heart, kidneys, testes, gastrointestinal tract, and lungs [24]. The ACE2 receptor is part of the renin–angiotensin–aldosterone system (RAAS) pathway and is vital in regulating inflammation and wound healing [25]. The interaction with the viral spike protein and the ACE2 receptor may result in a dysregulation of the RAAS pathway, reducing the levels of ACE2 and leading to increased angiotensin II production and inflammation [26]. The downregulation of ACE2 causes an increase in inflammatory cytokine release, also known as the cytokine storm [27]. The cytokine storm is an out-of-control production of inflammatory cytokines that leads to excessive inflammation in the lungs [28].

The viral spike S glycoprotein consists of two subunits: S1 and S2. The C-terminal receptor-binding domain of the S1 subunit acts on the recognition of the ACE2, while S2 subunit helps the fusion of viral membrane into host cell membrane. Following this binding, a transmembrane serine protease (TMPRSS2) or other proteases such as furin or cathepsin L catalyze the cleavage of the S protein [29]. TMPRSS2 is a cell surface protease and thus mediates the cleavage of the S protein at the plasma membrane. However, cathepsin L is located in the endosomal compartment and thus mediates the cleavage within the endolysosome. Cell entry by SARS-CoV-2, therefore, depends on the target-cell proteases, with TMPRSS2 and cathepsin L remaining as the two major proteases involved in S protein activation [30].

Additional mediators have been reported to facilitate virus entry in some cell types. These include non-tyrosine kinase, neuropilin (NRP1), the kidney injury molecule-1 (KIM1), the glucose-regulated protein 78 (GRP78), the metabotropic glutamate receptor subtype 2 (mGluR2), the heat shock protein A5 (HSPA5 or GRP78), the transmembrane glycoprotein CD147, and the G protein-coupled receptor mGluR2 (GRM2) [29,30]. Other molecules that may act as co-factors in the entry process include ADAM metallopeptidase domain 17 (ADAM17), which is involved the shedding of the ACE2 ectodomain, and heparan sulfate (HS), which mediates ACE2 viral entry [31]. Further strategies of infections incorporate specific binding sites such as O-linked or N-linked glycans on the outer membrane of SARS-CoV-2, and hence other host molecules such as sugars and sialic acids may act as potential virus receptors [32].

### 2.3. Host Cells Responses to SARS-CoV-2

The factors that trigger severe illness in patients infected with SARS-CoV-2 are not yet clearly defined, and the development of severe symptoms does not seem to be only related to the viral load or spread. Rather, severe disease appears to be associated with exaggerated host cell responses [33,34]. In this review, we present current knowledge that focuses on host cell responses related to SARS-CoV-2 infection, including immune and nonimmune cell responses. The ACE2 receptor is expressed in many host cell types, including type 2 alveolar cells, ciliated cells, and goblet cells in the airways, which provide a port of entry for the virus in humans that leads to infection of the intestinal epithelium, cardiac cells, and vascular endothelia (Figure 2) [35,36].

#### 2.3.1. SARS-CoV-2 in Immune Cells

An excessive inflammatory response to SARS-CoV-2 is thought to be a major cause of disease severity and death in patients with COVID-19 [37]. This is associated with high levels of circulating cytokines and substantial mononuclear cell infiltration in the lungs, heart, kidneys, and other organs [38,39,40]. A stronger understanding of the different immune cell responses to SARS-CoV-2 is necessary to identify better therapeutic strategies; in particular, the cell responses related to the excessive inflammation associated with COVID-19 and how it is correlated with high rates of morbidity and mortality [33]. COVID-19 severity and death has been associated with increased serum levels of several inflammatory cytokines and chemokines as well as an increased neutrophil:lymphocyte ratio [41,42]. The systemic cytokine profiles observed in patients with severe COVID-19 show similarity to those observed in activated macrophages, with increased production of IL-6 and IL-7, tumor necrosis factor (TNF), and the inflammatory chemokine CC–chemokine ligands CCL2 and CCL3 [43,44]. Monocyte/macrophages in COVID-19 bronchoalveolar fluid from patients with severe COVID-19 were shown to express CCL2 and CCL7 [45]. It is of interest that SARS-CoV-2 is not able to productively replicate in human primary macrophages and dendritic cells or in T and B cells, suggesting that inflammatory cytokine production in immune cells is triggered by other mechanisms that direct virus replication [17].

Macrophages that contained SARS-CoV-2 viral particles were found to express IL-6, while CD14+ and CD16+ monocytes, which specifically produce IL-6, were observed in the peripheral blood of patients with severe COVID-19 [46,47,48]. SARS-CoV-2 shares almost 80% of its sequence homology with SARS-CoV and 50% with MERS-CoV [49]. High levels of interferon-γ (IFNγ), IL-6, IL-8, IL-12, transforming growth factor-β (TGFβ), CCL2, CXCL9, and CXCL10 were reported in the blood of patients with SARS-CoV or SARS-CoV-2 infection [50,51]. In contrast to patients infected with SARS-CoV-2, low levels of IL-10 and high levels of IL-1β were detected in SARS-CoV-infected patients [33,34].

The host response and the clearance of viral infections rely generally on type I and III interferon (IFN) production [52]. Immune cells sense viral infection through identification of virus-derived pathogen-associated molecular patterns (PAMPs), such as viral RNA. These bind to and activate pattern recognition receptors (PRRs) in immune cells. RNA viruses such as SARS-CoV-2 are detected by PRRs, including the Toll-like receptors (TLR-)3 and 7 and/or the cytoplasmic retinoic acid-inducible gene I (RIG-I) receptors, RIG-I, and MDA5. This results in nuclear translocation of the transcription factors NFκB and IRF3, which triggers an increased expression of type I and III IFN and other proinflammatory cytokines (IL-1β, IL-6, and TNF-α) [53,54]. These immune responses should result in pathogen clearance and recovery, which does, however, seem to downregulated by SARS-CoV-2, likely contributing to a more severe disease and poor prognosis [34].

The exact mechanisms of monocyte and macrophage activation in COVID-19 remains to be clarified, raising questions regarding the exact contribution of direct viral sensing as opposed to cytokine exposure to macrophage activation [55], the influence of prior infections and epigenetic remodeling events in shaping monocyte responsiveness [41], the contribution of the tissue site of immune activation (that is, the inflamed tissue, blood, or bone marrow) to the macrophage activation state, and the contribution of tissue-resident macrophages as opposed to monocyte-derived macrophages to tissue damage [33].

Though seemingly contradictory to immune evasion mechanisms, enhanced innate immune activation, including systemic type I IFN, IL-1β, IL-6, and TNF-α production, contributes to morbidity and mortality in COVID-19 [56]. Marked changes are generally observed in the immune cell composition and phenotype in SARS-CoV-2 infection and the immunological features of severe COVID-19 in patients with ARDS [57]. It is of interest that SARS-CoV has the ability to circumvent the innate as well as the adaptive immune system mediated through dendritic cells (DCs). The transport of the virus to the lymphatic tissue by DCs followed by the infection of susceptible target cells might play a crucial role in the downregulation of the immune response in SARS patients [58].

In contrast to monocytes and macrophages, T-cell lymphopenia has been observed in many patients with severe COVID-19. The potential mechanisms responsible for T-cell depletion have not yet been studied, but are not likely related to direct infection of T cells by the virus [17,59,60]. Patients who died of COVID-19 showed an increased expression of the death receptor FAS, suggesting that activation-induced cell death may be related to T-cell lymphopenia [59,60].

#### 2.3.2. Endothelial Cells

The vascular endothelium is an active paracrine, endocrine, and autocrine organ that is responsible for the regulation of vascular tone and the maintenance of vascular homoeostasis [61]. Endothelial dysfunction is a principal determinant of microvascular dysfunction by shifting the vascular equilibrium toward vasoconstriction with subsequent organ ischemia, inflammation with associated tissue edema, and a procoagulant state [62]. Cardiovascular complications are rapidly emerging as one of the key disease mechanisms in COVID-19; however, the mechanisms involved are yet to be fully understood [63]. Patients are usually not dying because of hypoxemia. Often, the cause of death is cardiovascular, with high-sensitivity cardiac troponin I being a more reliable marker for mortality [64].

ACE2 receptors are also expressed by endothelial cells, and thus SARS-CoV-2 can directly infect engineered human blood vessel organoids and endothelial cells involved across vascular beds of different organs in COVID-19 patients [63]. In histological analyses, an accumulation of inflammatory cells—especially mononuclear cells—is associated with the endothelium, small bowel, and lungs, and the majority of small lung vessels appeared to be congested. Lymphocytic endotheliitis in the lungs, heart, kidneys, and the liver as well as liver cell necrosis were also observed [63]. All provided evidence suggested a direct viral infection of the endothelial cells and endothelial inflammation whether or not the recruitment of immune cells had occurred.

The endothelium is in much better condition in children compared to that in adults. Many adults with severe COVID-19 experience clotting of their blood vessels, which can further lead to heart attacks or strokes [65]. It was found that SARS-CoV-2 had infected the endothelium and caused inflammation and signs of clotting due to potentially disturbed communication between the cells, platelets, and plasma components involved in clotting [66], while it was observed that very few children with COVID-19 present conditions of excessive clotting and damaged vessels.

#### 2.3.3. Respiratory Tract Epithelial Cells

ACE2 is expressed in multiple epithelial cell types across the airway as well as in alveolar epithelial type II cells in the parenchyma, with the highest expression in nasal epithelial cells (including goblet cells and ciliated cells) [67,68,69,70,71]. The site of infection is thought to most likely occur in cells that express both ACE2 and the serine protease TMPRSS2; these sites include the corneal conjunctiva in the eye, ciliated and secretory cells in the nose, ciliated and secretory cells in the conducting airways, and alveolar type II cells in the gas exchange area of the lung [70]. No difference in SARS-CoV-2 receptor expression has been shown between symptomatic and asymptomatic patients’ nasal swabs that have yielded higher viral loads than throat swabs, implicating the nasal epithelium as a portal for initial infection and transmission [72].

In an interesting comparison, SARS-CoV-2’s replication competence was similar to that of MERS-CoV, higher than that of SARS-CoV, but lower than that of the influenza virus A/H1N1pdm09 in the bronchus. In the lungs, SARS-CoV-2’s replication was similar to those of SARS-CoV and A/H1N1pdm09 but was lower than that of MERS-CoV. In the conjunctiva, SARS-CoV-2 replicated to higher levels than SARS-CoV. SARS-CoV-2 was a less potent inducer of proinflammatory cytokines than A/H5N1, A/H1N1pdm09, or MERS-CoV [73].

Another cell type known as the club cell, which is located in the terminal respiratory bronchioles, may be an important site of infection based on its high expression of ACE2 [74,75]. The clinical picture of COVID-19 is mostly a mild infection of the upper respiratory tract where the infection originated and is located [76]. The disease in the gas exchange portions of the lung can be very severe and may take a long time to fully recover. As the host response to SARS-CoV-2 infection in the nose, the conducting airways, and the alveoli is usually milder, it is also likely that the innate immune response in these three sites are weaker. More information is needed regarding the characteristics of the infection and response in club cells, which have a restricted location in the human lung [76].

#### 2.3.4. Tubular Cells and Podocytes

It has been demonstrated that SARS-CoV-2 can infect podocytes and tubular epithelial cells, which also express ACE2. The infection of kidney cells very likely contributes to the development of renal abnormalities [77]. Acute kidney injury (AKI) is infrequent in the context of mild and moderate SARS-CoV-2 infection since functional abnormalities of the kidneys have been rare. Patients with moderate to severe symptoms of the disease showed different levels of proteinuria and hematuria and elevated levels of either serum creatinine, urate, or both [78]. It was found that SARS-CoV-2 antigens accumulated in renal epithelial tubules, suggesting that SARS-CoV-2 directly infects the kidneys, which may lead to AKI [79]. SARS-CoV-2 virus particles were also found in the proximal tubular epithelium and podocytes via electronic microscopy [80]. The effects of infection in podocytes have shown occasional vacuolation and detachment of the cells from the glomerular basement membrane [80]. These findings led to the hypothesis that proteinuria is a partial consequence of direct podocyte infection with potential alterations that affect the glomerular filtration rate and results in increased filtration of plasma proteins [78]. More information is needed to understand the interplay between kidney cells and other cell types like pericytes, the endothelium, interstitial cells, and immune cells that are involved in the establishment and maintenance of kidney dysfunction during COVID-19.

#### 2.3.5. Hepatocytes and Intestinal Cells

SARS-CoV-2 uses ACE2 receptors located on certain intestinal cells, cholangiocytes, and hepatocytes to infect the liver, leading to hepatic manifestations of the disease [81]. Cells of the gastrointestinal tract are likely also infected since diarrhea has been observed in up to 73% of patients within the first week of illness [82]. ACE2 is expressed in enterocytes of the ileum and colon [83,84,85]. ACE2 is partly responsible for mediating inflammation, which may contribute to the occurrence of diarrhea [54]. This has raised a question regarding whether or not SARS-CoV-2 is transmissible by the fecal–oral route [81,82,83,84,85]. Interestingly, in a recent Singaporean study, SARS-CoV-2 was detected in stool samples; however, this observation did not correlate with the symptoms of the gastrointestinal tract [86].

With regard to liver manifestations of COVID-19, studies have shown elevated levels of serum bilirubin, AST, and ALT in 10%, 21%, and 22% of patients, respectively [87,88]. Physicians have prescribed experimental drugs in COVID-19 patients showing liver disease even though most drugs are likely to be eliminated by the liver [81]. At present, the disease mechanisms contributing to gastrointestinal and hepatic manifestations of COVID-19 have remained elusive.

#### 2.3.6. Neural Cells

The onset of pneumonia and ARDS can occur rapidly in patients with COVID-19, suggesting a potential neuronal involvement in the disease pathology and mortality [89]. COVID-19 patients present a number of different neurological symptoms such as headache (most common), dizziness, hyposmia, cerebrovascular diseases, meningitis/encephalitis, acute necrotizing hemorrhagic encephalopathy, and acute Guillain–Barré syndrome [90]. Furthermore, SARS-CoV-2 RNA detection in the cerebrospinal fluid supports the idea of neurotropic involvement of SARS-CoV-2, although this event appears to be rare [90]. It was hypothesized that SARS-CoV-2 infection may increase the expression in the lungs of cytokines and chemokines, which then interact with receptors expressed by the sensory neuronal innervation of the lungs that may promote disease severity [90]. It has been reported that almost 40% of the patients have nonspecific neurological manifestations (e.g., dizziness, headache, and seizures) or specific manifestations such as dysfunction of smell or taste and strokes. It is unclear whether these symptoms are directly related to SARS-CoV-2 infection. However, neurologic symptoms such as a decreased level of consciousness, seizures, and strokes are relatively common in patients at late stages of the disease, accounting for increased mortality in severely infected patients [90,91].

Glial cells and neurons have been reported to express ACE2 receptors, which make them a potential target of SARS-CoV-2 infection [92]. It was also shown that SARS-CoV-2 may cause neuronal cell death by invading the brain via the olfactory epithelium [93]. The electron microscopy, immunohistochemistry, and PCR findings have corroborated the presence of SARS-CoV-2 in the brain and cerebrospinal fluid [93,94]. Overall, it can be speculated that SARS-CoV-2 can affect the brain through penetration via the cribriform plate, which can account for the early findings of COVID-19 such as an altered sense of smell or taste [92].

It has been demonstrated that ACE2 is expressed in neurons, astrocytes, oligodendrocytes, the substantia nigra, ventricles, the middle temporal gyrus, the posterior cingulate cortex, and the olfactory bulb [95]. In murine models, ACE2 expression has been identified in the motor cortex, cytoplasm of neurons, glial cells, and sympathetic pathways in the brainstem [96,97]. In neuronal cell cultures, ACE2 is expressed both on the cell surface and in the cytoplasm [98]. Viral neuroinvasion may be achieved via several routes, including trans-synaptic transfer across infected neurons, via the olfactory nerve, infection of the vascular endothelium, or leukocyte migration across the blood–brain barrier (BBB) [99]. One study underlies two mechanisms of SARS-CoV-2 and how the CNS may be infected (hematogenous dissemination or neuronal retrograde dissemination), whereas the indirect infection pathways have to be further elucidated in the future [90].

Through hematogenous dissemination, a virus can infect endothelial cells of the BBB to gain access or infect leukocytes for dissemination into the CNS [100,101,102,103]. SARS-CoV-2 may also bind to ACE2 expressed in the capillary endothelium of the BBB to gain access into the CNS [104,105]. Otherwise, SARS-CoV-2 can infect monocytes and macrophages to migrate through the BBB [105,106]. As a second mechanism, some viruses infect neurons in the periphery and use the axonal transport machinery to enter the CNS [100,101,102]. Another potential route for SARS-CoV-2 to enter the CNS is via the cranial nerve, as ACE2 is widely expressed on the epithelial cells of the oral mucosa. Moreover, olfactory receptor neurons project dendrites into the nasal cavity and extend axons through the cribriform plate into the olfactory bulb of the brain [107]. In addition to SARS-CoV-2 directly infecting the CNS, SARS-CoV-2 may indirectly affect the CNS through intracranial cytokine storms, resulting in the breakdown of the BBB without direct viral invasion [108,109].

## 3. Potential Drug Candidates and Clinical Trials for COVID-19 Treatment

Nucleoside analogs are RdRp inhibitors that can be used to treat different coronavirus infections, such as those caused by SARS and MERS [110,111]. The urgent demand for treatments has led the World Health Organization (WHO) to allow clinical trials of drugs that include: (a) hydroxychloroquine; (b) lopinavir; (c) ritonavir; (d) darunavir; and (e) remdesivir. Each have shown promising results for treatment of coronavirus infections [112]. However, the WHO and the FDA have halted most clinical trials due to lack of clinical evidence to support short-term recovery coupled with the inability to report possible long-term effects of emergency use of antimalarial and antiviral agents due to time constraints [113].

Several experimental clinical trials have concentrated on drug discovery and repurposing existing antiviral therapies, specifically those demonstrating prior effectiveness against MERS-CoV and SARS-CoV (Figure 3). Nafamostat, an inhibitor of MERS-CoV that prevents membrane fusion via the S protein, had an inhibitory effect against SARS-CoV-2 [114]. Notably, two compounds—remdesivir and chloroquine—inhibited viral infection at low-micromolar concentrations in vitro [8]. Each will be discussed in the sections below. Since the RdRp enzyme is highly conserved across all human coronaviruses, it is a prime target for inhibition by antiviral drugs, making remdesivir a front runner for COVID-19 treatment since it inhibits the RdRp activity [112].

### 3.1. Potential Candidates

#### 3.1.1. Antimalarial Agents

Chloroquine (CQ) and Hydroxychloroquine (HCQ): Both are amino acidotropic forms of quinine and are FDA-approved to treat or prevent malaria and autoimmune diseases [115]. They have shown to prevent the entry and transport of coronaviruses by increasing the endosomal pH required for virus and cell fusion and interfering with the glycosylation of cell receptors [116]. CQ and HCQ were expected to prevent the replication and invasion of the virus and inhibit T-cell activation, reducing the risk of a cytokine storm [117]. On 15 June 2020, the FDA determined that CQ and HCQ are unlikely to be effective in treating COVID-19 due to serious cardiac adverse events and other serious side effects, concluding that these medicines showed no benefit for decreasing the likelihood of death or expediting recovery for hospitalized patients and ultimately revoking the emergency use of CQ and HCQ [118].

#### 3.1.2. Antiviral Agents

Remdesivir (GS-5734) is an adenosine triphosphate nucleoside analog that interferes with virus replication and has been recognized as a promising antiviral drug against a plethora of RNA viruses, including Ebola, SARS, and MERS. However, its safety and efficacy are currently being tested in multiple clinical trials [119,120]. Remdesivir has been reported to inhibit viral RdRp activity at early stages of the infection cycle, and the incorporation of its active triphosphate form by the SARS-CoV-2 RdRp complex is higher than that of the competing natural ATP substrate, disrupting replication of the virus [121]. In June 2020, the FDA approved remdesivir for the treatment of COVID-19 requiring hospitalization. This approval was supported by analysis of data from three randomized, controlled clinical trials that included patients hospitalized with mild to severe COVID-19 [122,123].

Ribavirin is a guanosine nucleoside analog known as a broad-spectrum antiviral that is effective against HIV, hepatitis C, herpes viruses, and other viruses [124,125]. Ribavirin interferes with RNA polymerase and RNA capping, interrupting the viral cells’ ability to replicate, causing recognizable degradation. Treatment plans typically combine ribavirin with interferon-α2a or β1 [124]. Ribavirin monotherapy for MERS has shown little positive effect on patients with severe symptoms. There was hope that combination therapy of ribavirin with lopinavir/ritonavir against hCoV infections would be effective [126,127]. A phase II trial in March 2020 found that such a triple therapy was successful at early stages of COVID-19 in reducing the length of hospital treatment [128]. However, in July 2020, the NIH withdrew support of the use of ribavirin treatment as it did not reduce COVID-19 mortality [129].

Interferons (IFNs) are secreted antiviral cytokines that are produced by virus-infected cells. IFNs are broad-spectrum antivirals with *α*, *β*, and λ subtypes that have been used for the treatment of hepatitis and have been previously tested to treat MERS and SARS-CoV [130]. However, trials for IFN therapy (*α* or *β*) for hCoVs have shown variable results. Two clinical trials for SARS patients with IFN treatment determined that while IFN-α was able to inhibit SARS replication, IFN-β had similar results when used as a combination therapy [131]. Another study deemed IFN-β1b the preferred treatment for MERS and IFN-α2b/ribavirin for SARS, while several trials suggested that IFN-α2b/ribavirin could also be effective against MERS [132,133]. In a large randomized controlled trial of hospitalized patients with COVID-19, the combination of IFN-β1a plus remdesivir showed no clinical benefit when compared to remdesivir alone [134,135]. Trials evaluating IFN-α were not sufficient to determine whether this agent provides a clinical benefit for patients [136,137].

Lopinavir/Ritonavir (LPV/RTV) are antiviral agents that act as HIV protease inhibitors that have been used for the treatment of SARS and MERS [138]. Lopinavir is commonly combined with ritonavir to increase its half-life and reduce side effects [139]. LPV and RTV along with interferon-β1 and ribavirin have shown greater effects in inhibiting coronavirus replication as compared to LPV or RTV alone [140]. Due to the lack of efficacy and increased incidence of adverse events, the clinical use of LPV/r in hospitalized COVID-19 patients was not recommended [141].

Umifenovir (Arbidol), also known as ARB, is an antiviral drug that can inhibit several DNA and RNA viruses, including the West Nile virus, Zika virus, and hepatitis C virus. Simulations showed that umifenovir inserts into the spike protein of the virus, blocking viral fusion to the cell [142]. Reaching phase IV of clinical trials, angiotensin II receptor blockers (ARBs) and ACE inhibitors have been used to help in treating moderate to severe infections in patients with pneumonia [143]. Another study showed that ARB combined with LPV/r was better than lopinavir alone [144]. However, arbidol has shown little to no effect on patients with mild or moderate symptoms of COVID-19 [145,146].

Favipiravir (Avigan/T-705) is a pyridinecarboxamide that was initially developed as an anti-influenza substance and is classified as an antiviral drug that selectively inhibits the RNA polymerase of RNA viruses [147]. Favipiravir was shown to be effective against the influenza strains A(H1N1) and A(H7N9), and it later was used as a potential treatment for the Ebola virus, West Nile viruses, arenaviruses, and other viruses [148]. Once inside the cell, favipiravir is ribosylated and phosphorylated into its active form, T-705RTP [149]. Enzyme kinetic analysis demonstrated that T-705RTP inhibited the incorporation of ATP and GTP in a competitive manner, preventing the strand extension of the virus [150]. Initial trials for use in response to COVID-19 suggested that favipiravir administration in individuals with mild to moderate infection has a strong potential to improve clinical outcomes [151]. However, recent studies showed that favipiravir does not improve clinical response in all COVID-19 patients admitted to hospital, and further high-quality studies of antiviral agents and their potential treatment combinations are warranted [152].

Gemcitabine (Gemzar): Gemcitabine hydrochloride is a pyrimidine nucleoside analog that is commonly used as a chemotherapeutic agent against several forms of cancer, such as pancreatic cancer [153]. Previous investigations of the effect of gemcitabine on MERS and SARS reported that the drug inhibits both HCoVs at a low micromolar EC_50_ of 1.22 and 4.96 μM, respectively [154]. One report suggested that the difluoro group of gemcitabine is critical to its antiviral activity and could be a desirable option to treat SARS-CoV-2 in combination with other antiviral drugs, such as remdesivir [155].

Nafamostat Mesylate (Fusan) is a broad-spectrum serine protease inhibitor that can act as an anticoagulant and is used to treat pancreatitis and other inflammatory diseases [156]. In 2016, nafamostat mesylate was identified as a treatment option for MERS, successfully inhibiting the host cell’s TMPRSS2 receptor during viral S protein–cell membrane fusion, setting it up for potential inhibition of other HCoVs [113]. In April 2020, when comparing serine protease inhibitors, one study showed that nafamostat mesylate had a 15-fold increase when compared with camostat mesylate in inhibiting SARS-CoV-2 S protein fusion mediated by the host cell ACE2/TMPRSS2 receptor complex [157]. Moreover, a case study reported the efficiency of both favipiravir and nafamostat mesylate as a joint treatment in SARS-CoV-2 patients, noting that the anti-blood clotting properties of nafamostat mesylate were likely beneficial in treating SARS-CoV-2 patients [158]. Additional clinical trials are needed to provide more robust data on the safety and efficacy of nafamostat as a treatment for COVID-19 [159].

Sofosbuvir/Daclatasvir: Both drugs are classified as direct-acting antivirals (DAAs) and are FDA-approved to treat the hepatitis C virus (HCV) due to their combined ability to inhibit the functions of HCV nonstructural protein 5 (NS5B), an essential component in viral replication [160]. Like HCV, SARS-CoV-2 is a positive-stranded RNA virus, making it the potential primary target of sofosbuvir/daclatasvir [161]. However, no significant reduction in the SARS-CoV-2 viral load was observed in patients hospitalized with COVID-19 and receiving this treatment plan when compared with a control group [162]. Larger clinical trials are warranted.

EIDD-2801 (molnupiravir) is an orally bioavailable ribonucleoside analog of β-D-*N*^4^-hydroxycytidine (NHC, EIDD-1931) [163]. NHC has been found to participate in viral replication by causing mutations from G to A and C to U, which is deadly to the new replication of the virus [164]. Studies have shown that EIDD-2801 is highly effective against SARS-CoV in Vero cells (IC_50_ of 0.3 µM) and in mice infected with MERS, and it was also effective against SARS-CoV-2 RdRp mutations that developed as a result of remdesivir treatment [165]. In November 2021, the U.K. became the first country to approve this drug for the treatment against mild and moderate SARS-CoV-2. Within a month, the U.S. FDA approved EIDD-2801 for emergency use [166]. EIDD-2801 is the first oral antiviral drug that demonstrated efficacy in decreasing the viral RNA amounts in nasopharyngeal swabs [167]. Relatedly, recent studies provide evidence of the efficacy of molnupiravir, which compliments the ongoing clinical trials of EIDD-2801 [168].

Galidesivir (BCX4430) is an adenosine nucleoside analog shown to have effective antiviral properties against several RNA viruses, including the Zika virus, HCV, and Ebola virus [169]. Two studies showed that galidesivir has an ability to inhibit SARS-CoV-2 RdRp and thus has potential as a therapeutic agent [170,171]. In cells where phosphorylation occurs, premature chain termination can be achieved. Structural modeling shows a potential allosteric inhibition of RdRp as galidesivir binds to the noncatalytic site of the enzyme [172]; however, early stages of clinical trials showed no benefit for COVID-19 patients [173].

#### 3.1.3. Other Agents and Therapies

Vitamin C: Studies have shown that intravenous (IV) vitamin C may help COVID-19 patients by reducing inflammation or possibly by restoring the antioxidant protection of the body [174]. Overall, evidence from randomized controlled trials suggests a survival benefit for vitamin C in patients with severe COVID-19; however, more data with definitive outcomes are needed before it is recommended to provide high-dose vitamin C therapy to prevent or treat COVID-19 [175]. It is currently advised to maintain a normal physiologic range of plasma vitamin C through diet or supplements for adequate prophylactic protection against the virus [176].

Corticosteroids are synthetic analogs of steroid hormones naturally produced by the adrenal cortex that have broad anti-inflammatory activities [177]. Dexamethasone is a fluorinated corticosteroid that relieves inflammation and is commonly used to treat arthritis [178]. In a clinical trial of hospitalized patients with COVID-19, the use of dexamethasone resulted in lower mortality among those receiving either mechanical ventilation or oxygen [179]. The WHO announced that only patients with severe or critical symptoms of COVID-19 should be administered dexamethasone (or other corticosteroids such as prednisolone and hydrocortisone), reiterating that mild cases do not benefit from the treatment [180]. This evidence comes from combined results of clinical trials that showed that deaths were reduced by 20% when patients with severe COVID-19 were given corticosteroid treatment during their illness, thus confirming the efficacy of corticosteroids during the hyperinflammatory stage of SARS-CoV-2 infection [181].

Bamlanivimab and etesevimab are monoclonal antibodies that target the receptor binding domain of the SARS-CoV-2 spike protein and thus prevent the entry of the virus into human cells [182]. Identifying the specific antibodies from a patient that recovered from SARS-CoV-2 infection expedited the development of this treatment in eight months [183]. Patients with mild or moderate symptoms were found to have reduced amounts of the virus in nasopharyngeal swabs [183]. Due to the increase in variants that are not susceptible to bamlanivimab, this product administered alone is not currently authorized by the FDA, with the prior EUA authorization now revoked as of April 2021 [184]. In September 2021, the FDA revised the emergency use authorization of bamlanivimab and etesevimab as a combination therapy [185]. The revision included emergency use of post-exposure prevention of prophylaxis for children that are 12 years of age and older. These monoclonal antibodies are not offered for pre-exposure but show positive results in treating patients that are at high risk for developing severe COVID-19 [186].

#### 3.1.4. Antibiotic/Antibacterial

Azithromycin (AZM) is a macrolide antibiotic that has been approved by the FDA to treat enteric and respiratory tract bacterial infections. It was investigated in combination therapy with hydroxychloroquine (HCQ) to treat hospitalized COVID-19 patients with bacterial infections [187]. A clinical trial of HCQ and AZM consisting of 504 COVID-19 patients randomized at a 1:1:1 ratio compared the outcomes of one group receiving standard care, another group treated only with HCQ, and the third group receiving an AZM/HCQ mixture. The results showed that the use of HCQ with or without AZM did not improve clinical status at 15 days when compared to the standard care treatment [188]. Several other studies with AZM as a therapy showed no increase in clinical benefit and that there was potential risk for development of antibacterial resistance [189]. The results from the COALITION II Trial, RECOVERY Trial, and PRINCIPLE trial showed that AZM was not better than standard care alone and did not provide any benefits in recovery. Following these findings, the NIH and WHO recommended against the use of antibacterial medicine for COVID-19 patients [190].

Cefuroxime, a broad-spectrum antibiotic that is commonly used in the treatment of respiratory and gastrointestinal tract infections, has been investigated as anti-SARS-CoV-2 agent [191]. In silico studies reported the antibiotic as a strong inhibitor against SARS-CoV-2’s major protease [192,193,194]. As such, these in silico studies showed that cefuroxime could potentially be successful in clinical trials due to its bioavailability and viral inhibitory activity. However, the gratuitous use of antibiotics for COVID-19 treatment without proper clinical rationale raised concerns about the global problem of the emergence of antimicrobial resistance [195].

Teicoplanin is a glycopeptide antibiotic used to treat Gram-positive bacterial infections [196] that has been of interest in the drug-repurposing efforts against SARS-CoV-2. Teicoplanin and its derivatives were shown to inhibit cathepsin L activity in Ebola, SARS, and MERS infections. Teicoplanin does not block cellular viral receptors or target viral particles, but its inhibitory action against cathepsin L leads to reduced low-pH-associated cleavage of the SARS-CoV S protein, preventing the S glycoprotein trimer from fusing with the host cell membrane [197]. In addition to its ability to reduce viral entry, teicoplanin possesses a lower toxicity and nephrotoxicity than the similar glycopeptide vancomycin, making it a preference for at-risk patients and those with pre-existing medical conditions [198]. More importantly, teicoplanin derivatives were reported to inhibit the entry of both pseudo-typed SARS-CoV-2 Delta and Omicron variants by targeting the interaction of the viral S protein and ACE2, suggesting them as promising pan-SARS-CoV-2 inhibitors [199].

#### 3.1.5. ACE2 Inhibitors

Captopril, a high blood pressure medication primarily marketed for patients affected by diabetes, congestive heart failure, or hypertension [200], was selected as a potential treatment against SARS-CoV-2 infection due to its angiotensin-converting enzyme (ACE) inhibitory action [201]. Experimental data suggested that captopril increased the level of ACE2 expression in lung cells, and this up-regulation was counteracted by drug-induced mechanisms that reduced SARS-CoV-2 spike protein entry [202]. The ACE2-centered therapeutic approaches to prevent and treat COVID-19 now need to be tested in clinical trials to combat current cases of COVID-19, including all SARS-CoV-2 variants and other emerging zoonotic coronaviruses exploiting ACE2 as their cellular receptor [203].

hrsACE2/rhACE2c: Human recombinant soluble ACE2 (hrsACE2) is a novel compound listed for clinical trials to test its efficacy in regulating a patient’s imbalance in the renin–angiotensin system (RAS) [204]. While one report that hrsACE2 successfully prevented SARS-CoV-2 entry and decreased viral RNA levels suggests that hrsACE2 might provide an effective SARS-CoV-2 treatment, further clinical trial data are needed to validate this finding [205].

Ramipril is an antihypertensive drug that has been considered as a potential treatment against SARS-CoV-2 [206]. Ramipril works by preventing counter-regulatory ACE from cleaving decapeptide angiotensin I into angiotensin II. Angiotensin II plays an important role in the renin–angiotensin–aldosterone system (RAAS), which regulates blood pressure, vasoconstriction, and extracellular volume [207]. Assessment of hypertensive drugs has been conducted extensively, and guidance for the use of ACE and ARBs (hypertensive drugs that operate in the same biochemical pathway as ACE2, i.e., the RAAS system) suggested that the use of ACEIs/ARBs did not significantly influence either mortality or severity in comparison to not taking ACEIs/ARBs in COVID-19-positive patients [208,209].

#### 3.1.6. Anti-Inflammatories

COVID-19 can trigger a cytokine storm through hyperactivation of the immune system and the uncontrolled release of cytokines. This uncontrolled immune response can lead to severe tissue damage and contribute to the progression of the disease. Thus, there has been significant interest in strategies to modulate the immune response in COVID-19 cases, with a focus on suppressing the cytokine storm. This can be achieved through the use of immunomodulatory drugs like corticosteroids or specific drugs that target cytokines or their receptors. However, it is important to note that while these drugs can be effective in managing such conditions, they also suppress the immune system, which can make patients more susceptible to infections. Therefore, patients receiving this medication are typically carefully monitored for signs of infections while on treatment.

Tocilizumab, which is marketed under the brand name Actemra, is an immunosuppressive humanized monoclonal antibody drug. It is used to treat a range of inflammatory conditions, including rheumatoid arthritis and juvenile idiopathic arthritis [210,211]. Tocilizumab is the first FDA-approved monoclonal antibody for treating patients with severe COVID-19 [212]. It works by targeting and inhibiting the activity of the interleukin-6 receptor (IL-6), a cytokine that plays a key role in various autoimmune and inflammatory diseases [213]. By blocking the IL-6 receptor, tocilizumab helps reduce the inflammatory response and can provide relief to patients. Tocilizumab has shown to decrease the duration of hospitalization, the risk of being placed on mechanical ventilation, and the risk of death for patients with severe COVID-19 [214].

Anakinra is an immunomodulatory drug that has been studied for its potential to modulate the immune response in COVID-19 cases. It antagonizes inflammation mediated by interleukin-1 (IL-1) via binding to the corresponding receptors, thereby preventing the cascade of sterile inflammation seen in various pathological states. Additionally, anakinra interferes with the assembly of the inflammasome, a multiprotein complex that plays a key role in initiating and regulating the inflammatory response [215]. Studies showed that anakinra might be beneficial in the early phase of inflammation in patients at risk for progression to respiratory failure; however, its effectiveness in patients already suffering from respiratory failure is not suggested [216]. A recent systematic review and meta-analysis concluded that anakinra has no effect on adult hospitalized patients with SARS-CoV-2 infection regarding mortality, clinical improvement, and worsening or on safety outcomes compared to a placebo or standard care alone [217].

Table 3 includes the most current therapeutics with known pharmacological properties to treat the SARS-CoV-2 coronavirus. It is critical for researchers to continue evaluating these molecules to propose drugs that could be effective not only for treatment of COVID-19 but also for the new variants and subvariants that continue to arise. Throughout the pandemic, the spike protein has consistently mutated, introducing new variants that the current treatments can no longer remedy. Of concern is the wide array of treatment candidates (such as remdesivir) that were previously authorized for emergency use, then revoked, but remain authorized to date. The treatment candidates presented in this paper provide insight and lessons for future researchers in the design of new therapeutics based on the failure or success of their predecessors, demonstrating the importance of reviewing past studies to better understand the virus and narrow the possibilities for efficacious treatment.

## 4. Pathways as Potential Targets

At present, researchers have primarily focused on treating SARS-CoV-2 infection by either blocking viral entry through the spike protein or inhibiting virus replication through the RdRp enzyme or the main protease (Nsp5) [218]. Identifying the mechanism(s) of internalization of SARS-CoV-2 to potentially inhibit this phase of the virus’s life cycle is of equal importance.

### Clathrin-Mediated Endocytosis

Clathrin is a key component in vesicle formation and transport that is involved in the coating of a vesicle during endocytosis [219]. Researchers found that after SARS-CoV-2 binds to ACE2, the virus enters the host cell via clathrin-mediated endocytosis [220]. Drugs such as chlorpromazine, ikarugamycin, and dynasore that inhibit clathrin-mediated endocytosis or membrane fusion may show anti-SARS-CoV-2 activity [221]. Chlorpromazine (CPZ) is an antipsychotic that is used for disorders such as schizophrenia. This drug is in phase III of clinical trials to treat COVID-19. CPZ is inhibiting clathrin-mediated endocytosis, and it has shown to have antiviral activity against SARS-CoV; it thus could also have potential as a treatment for SARS-CoV-2 [221]. Ikarugamycin (IKA) is an antibiotic that has previously been used to inhibit the clathrin-mediated endocytosis pathway [220,221]. Further studies are needed to establish the efficacy of this antibiotic on COVID-19. Dynasore is a GTPase inhibitor that acts on dynamin. Dynamin is essential for membrane fission during clathrin-mediated endocytosis [220,221]. Targeting the clathrin pathway with various compounds may prove effective in interfering with the virus entry as well as with the budding of a newly formed viral particles. There are several drugs that can target this important pathway, and the clinical activity of these compounds against COVID-19 should be analyzed.

## 5. Computer-Aided Drug Design

To combat COVID-19, bioinformaticians around the world have reacted quickly by providing free and available online tools designed to accelerate SARS-CoV-2 research. Computational strategies have been developed for the routine detection of SARS-CoV-2 infection, the tracking of the COVID-19 pandemic, the study of coronavirus evolution, the detection of drug targets, and potential treatments [222]. Rational drug design strategies have been investigated during the COVID-19 pandemic [223]. This includes structural design of molecules that could bind to the spike protein [224] or ACE2 [225] to inhibit virus interaction with its cellular receptor. Additionally, molecules and nucleoside analogs that can bind to viral proteases and RdRp were reported [226,227]. Repurposing antiviral and anticancer drugs, namely nucleoside analogs, have proven successful at viral inhibition, which we have demonstrated throughout this review.

## 6. Conclusions and Perspectives

COVID-19 has a broad clinical spectrum; in addition to the main infection of the lungs, SARS-CoV-2 may also affect other organs, including the gastrointestinal system, liver, kidneys, and heart or blood vessels. The widespread expression of ACE2, the receptor that allows SARS-CoV-2 to bind its target host cell, contributes to viral spread into different organs. It is clear that hyperinflammation and coagulation disturbances contribute to disease severity and death in patients infected with SARS-CoV-2. The research articles included in this review have heightened the awareness of effective antivirals against COVID-19 and potentially the previous human coronaviruses as well as COVID-19 variants. In addition to this approach, several efforts to design, synthesize, or repurpose effective therapeutics have led companies to race for new treatment options. As a result, many clinical trials have been conducted and expedited to meet current demands for a treatment.

Researchers continue to pose several questions with very broad directions that still require answers. The spectrum of these questions varies, and researchers are investigating, for instance, how this novel coronavirus dysregulates host immune responses. It is of special interest that some people can become re-infected shortly after recovering from an initial COVID-19 infection [228]. It is still unclear how humoral and cell-mediated immunity contributes to the susceptibility to and severity of infection. The appearance of new virus variants that show increased transmissibility and the circumvention of pre-existing immunity clearly contribute to new epidemic waves of COVID-19. Novel and repurposed antiviral drugs that effectively inhibit virus replication are urgently needed for infected individuals. Additionally, further information and the means to control the pathogenic inflammation in severe COVID-19 is critical.

We have presented the way in which the FDA has authorized a plethora of drugs for the treatment of mild to severe COVID-19, with several having now been revoked due to studies demonstrating toxicity, no reduction in mortality, etc. The cessation of COVID-19 as a public health emergency (PHE) will not impact the ability of the Center for Drug Evaluation and Research (CDER) to authorize treatments for emergency use, and the existing emergency use authorizations (EUAs) remain in effect. Three critical questions remain: whether treatment with ACE2 inhibitors or receptor blockers during active COVID-19 would be of value during the course of infection; whether patients with chronic diseases (who have higher expression of ACE2) have a higher risk of infections or disease progression; and whether children have a lower level of this receptor expressed in their healthy vessels. Finding answers to these questions is crucial to developing better therapeutic possibilities.

This review contributes to future research by providing a process by which researchers designing new therapeutics can review the failure or success of a candidate’s predecessors to better understand an emergent virus, thereby narrowing the possibilities for effective treatment while expediting efficacious patient care.

## Figures and Tables

**Figure 1 biomolecules-13-01452-f001:**
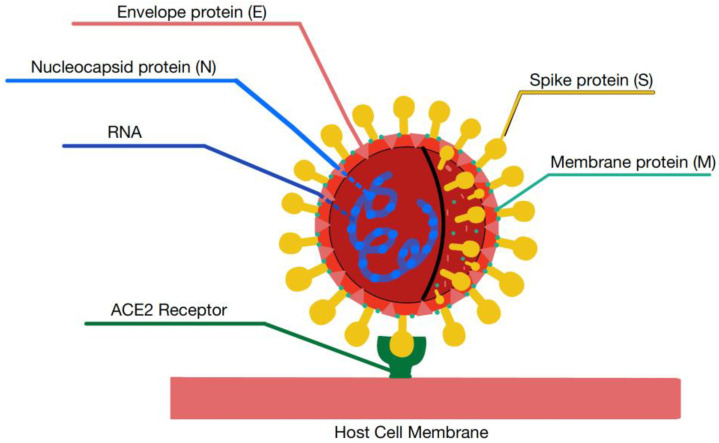
Structure of SARS-CoV-2 and its structural proteins.

**Figure 2 biomolecules-13-01452-f002:**
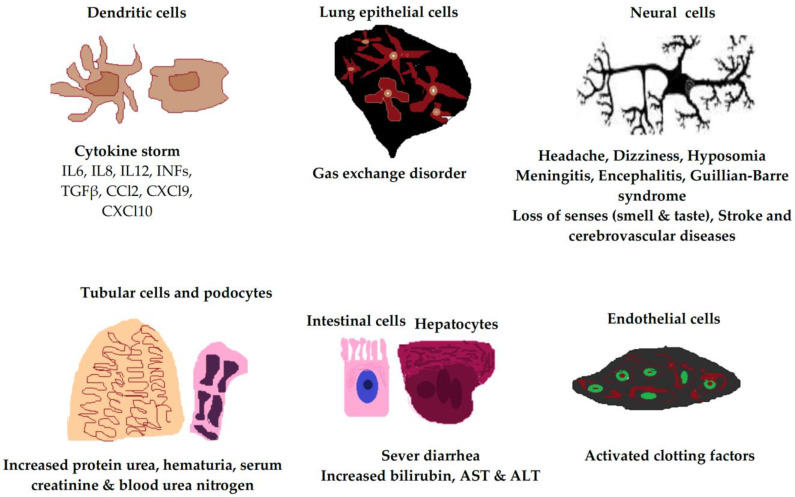
Clinical manifestations of SARS-CoV-2 infection in different human cell types due to ACE2 receptor expression. The ACE2 receptor is expressed in most cell types of the body, including neural cells, alveolar epithelial cells, immune cells, vascular endothelial cells, intestinal cells, hepatocytes, tubular cells, and podocytes. IL, interleukin; INF, interferon; TGFβ, transforming growth factor beta; AST, aspartate transaminase; ALT, alanine transaminase.

**Figure 3 biomolecules-13-01452-f003:**
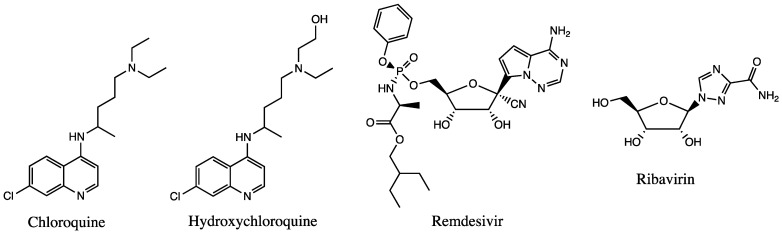
Chemical structures of available drugs that have been investigated for the treatment of COVID-19.

**Table 1 biomolecules-13-01452-t001:** History of coronaviruses and their classification as *alpha*-, *beta*-, *gamma*-, or *delta*-CoVs [3,4].

Year	Virus Name	Genera	Host/Origin/Intermediate Origin	Cell Receptor	Symptoms
1952	TGEV	α-CoVs	Pig	Porcine APN	Mild Respiratory tract infection and Enteric infection
1966	HCoV-229E		Human/Bat/Alpaca	Human APN	Mild Respiratory tract infection
1988	HCoV-NL63		Human/Bat/Unkown	ACE2	Mild Respiratory tract infection
1961	MHV-A59	β-CoVs	Mouse	Murine CEACAMI	Acute pneumonia and severe lung injuries
1967	HcoV-OC43		Human/Bat/Cow	Neu5, 9Ac-2-containing moiety	Mild Respiratory tract infection
2003	SARS-CoV		Human/Bat/Palm Civets	ACE2	Severe acute respiratory syndrome, 10% mortality
2004	HKU-1		Human/Mouse/Unkown	HLA	Pneumonia
2008	CCoV		Dog/Dog	Canine APN	Diarrhea, Enteric infection
2012	MERS-CoV		Human/Bat/Camel	CD26 (DPP4)	Severe acute respiratory syndrome, 37% mortality
2015	Bat-SI, CoVZC21		Bat/Bat	n/a	n/a
2017	Bat-SI, CoVZC45		Bat/Bat	n/a	n/a
2019	SARS-CoV-2		Human/Bat/Unkown	ACE2	Severe lower respiratory tract infection
1935	IBV	γ-CoVs	Chicken/Avian	S glycoprotein	Severe respiratory disease
2008	SW1		Whale	n/a	Pulmonary disease, terminal acute liver failure
2007	HKU11	δ-CoVs	Bulbul/Pycocotus/Jocosus	n/a	Respiratory disease collected from respiratory tract of dead birds
2007	HKU17		Sparrow/Passer/Montanus	n/a	Respiratory disease collected from respiratory tract of dead birds

APN, aminopeptidase N; ACE2, angiotensin-converting enzyme 2; HLA, human leukocyte antigen; DPP4, dipeptidyl peptidase 4.

**Table 2 biomolecules-13-01452-t002:** Nonstructural proteins and their functions [3].

Nonstructural Proteins	Functions
Nsp1	Degradation of cellular mRNA, inhibition of interferon (IFN) signaling
Nsp2	Unknown
Nsp3	Cleavage of PLP polypeptides, inhibition of IFN signaling
Nsp4	Formation of DMV
Nsp5	Cleavage of 3CLpro, Mpro, polypeptides, inhibition of IFN signaling
Nsp6	Restriction of autophagosome expansion, formation of DMV
Nsp7	Co-factor of holo-RdRp
Nsp8	Co-factor of holo-RdRp
Nsp9	Capping and binding of RNA
Nsp10	Scaffold protein for nsp14 and nsp16
Nsp11	Unknown
Nsp12	Primer-dependent RdRp
Nsp13	RNA helicase, 5′-triphosphatase
Nsp14	Exoribonuclease, N7 Mtase
Nsp15	Endoribonuclease, evasion of dsRNA sensors
Nsp16	RNA-cap-2′-O-methyltransferase, inhibition of MDA5 recognition, negative regulation of innate immunity

**Table 3 biomolecules-13-01452-t003:** Summary of therapeutics with known pharmacological properties to treat the SARS-CoV-2 coronavirus. The ending of the COVID-19 public health emergency (PHE) will not impact the ability of the Center for Drug Evaluation and Research (CDER) to authorize treatments for emergency use, and the existing emergency use authorizations (EUAs) for products remain in effect.

Drug	Classification	Target	Treatment of/Usage	FDA Approved for COVID-19	EUA for COVID-19	Comments
Actemra (Tocilizumab)	Monoclonal Antibody	Interleukin-6 receptor antagonist	Arthritis, SSc-Interstitial lung disease, Cytokine Release Syndrome	Yes		for adults receiving systemic corticosteroids and require supplemental oxygen, non-invasive/invasive mechanical ventilation, or extracorporeal membrane oxygenation
Veklury (Remdesivir)	Antiviral	RdRp inhibitor	Ebola, SARS, and MERS	Yes		for adults and pediatric patients (with age and weight limitation), have mild-to-moderate and high risk for progression to severe COVID-19
Olumiant (Baricitinib)	Immune Modulator	Janus kinase inhibitor	Rheumatoid arthritis	Yes		for hospitalized adults requiring supplemental oxygen, non-invasive/invasive mechanical ventilation, or extracorporeal membrane oxygenation
Paxlovid (Nirmatrelvir & Ritonavir)	Antiviral	Protease inhibitor, CYP3A inhibitor	HIV	No	Yes	for adults and pediatric patients (with age and weight limitation), have mild-to-moderate and high risk for progression to severe COVID-19
Lagevrio (Molnupiravir)	Antiviral	Viral mutagenesis	Influenza	No	Yes	for adults who have mild-to- moderate and high risk for progression to severe COVID-19
Kineret (Anakinra)	Immune Modulator	Interleukin-1 receptor antagonist	Rheumatoid Arthritis, Cryopyrin-Associated Periodic Syndromes, Deficiency of Interleukin-1 receptor antagonist	No	Yes	for adults with pneumonia requiring supplemental oxygen and at risk for of progressing to severe respiratory failure
Gohibic (Vilobelimab)	Immune Modulator	C5a receptor blocker	n/a	No	Yes	for adults when initiated within 48 h of receiving invasive mechanical ventilation or extracorporeal membrane oxygenation
REGEN-COV (Casirivimab & Imdevimab)	Monoclonal Antibody	SARS-CoV-2 spike protein binding domain receptor	n/a	No	Yes	for adults and pediatric patients (with age and weight limitation), have mild-to-moderate and high risk for progression to severe COVID-19
Sotrovimab	Monoclonal Antibody	SARS-CoV-2 spike protein binding domain receptor	n/a	No	Yes	for adults and pediatric patients (with age and weight limitation), have mild-to-moderate and high risk for progression to severe COVID-19
* Bamlanivimab & Etesevimab	Monoclonal Antibody	SARS-CoV-2 spike protein binding domain receptor	n/a	No	Yes	for adults and pediatric patients (with age and weight limitation), have mild-to-moderate and high risk for progression to severe COVID-19
Bebtelovimab	Monoclonal Antibody	SARS-CoV-2 spike protein binding domain receptor	n/a	No	Yes	for adults and pediatric patients (with age and weight limitation), have mild-to-moderate and high risk for progression to severe COVID-19
Evusheld (Tixagevimab with Cilgavimab)	Monoclonal Antibody	SARS-CoV-2 spike protein binding domain receptor	n/a	No	Yes	for certain adults and pediatric patients as pre-exposure prophylaxis of COVID-19

* SARS-CoV-2-targeting monoclonal antibodies (mAbs) are laboratory-produced antibodies that block the viral entry into human cells. SARS-CoV-2 can mutate over time, resulting in variants that are resistant to one or more mAbs therapies authorized to treat COVID-19. Due to the increase in variants that are not susceptible to bamlanivimab, this product administered alone is not currently authorized by FDA and the EUA previously authorized has been revoked in April 2021.

## Data Availability

Not applicable.

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
