# Peer review of "Insights into COVID-19: Perspectives on Drug Remedies and Host Cell Responses"

_biomolecules, 2023, doi:10.3390/biom13101452_

Round 1

Reviewer 1 Report (Previous Reviewer 3)

Insights into COVID-19: Perspectives in drug remedy and host cell responses” by Awad et al. is a review of SARS-CoV-2. The paper is logically structured with a history of coronaviruses, structural components of the virus, entry into human cells and replication, and pathology experienced in infected individuals.  Next, repurposing existing drugs to treat COVID-19 and improve outcomes is described, and often highlights clinical trials in patients with COVID-19.  Finally, the authors outline their development of new anti-viral agents with their preliminary results. While understanding COVID-19 biology and therapeutic interventions is important, this review has substantial shortcomings.

1: The review is not novel.  A search of PubMed for “COVID-19 AND Therapeutics AND Review” returns 1827 papers published 2020-2023. 

2: There are glaring omissions on the review, including but not limited to:

-          Long COVID.

-          What was learned from failed versus successful drugs?  Without this, discussion of failed drugs is extraneous, and there’s no lessons for the next generation of COVID-19 drugs.

-          CYP3A and C5a are drug targets on new Table 3 that are simply missing from the review.

3: Hyper-inflammatory responses are the leading cause of morbidity and mortality from COVID-19, and so therapeutics targeting inflammation have been evaluated far beyond what this review includes. The new section on anti-inflammatory therapeutics to improve outcomes is inadequate (Section 3.1.6).  It does not relate therapeutic interventions to inflammatory processes in Section 2. It also ignores guidance related to NSAIDs or any other anti-inflammatory, provides essentially no molecular mechanism (all glaring omissions), and includes vaccines - which are not anti-inflammatories.  Even their new Table 3 includes anti-inflammatory therapeutics which are not discussed e.g. Actemra (Tocilizumab) and Kineret (Anakinra).

4: Inaccurate information:

Section 3.1.5 inaccurately describes Captopril as an ACE2 inhibitor, when it is an ACE inhibitor.

Meanwhile, the discussion on Ramipril, also an ACEi, states clinical studies for use of hypertensive medications in patients with COVID-19 are warranted. This is not accurate.  Assessment of hypertensive drugs has been done extensively, and guidance for use of ACEi and ARBs (hypertensive drugs that operate in the same biochemical pathway as ACE2 i.e. RAAS system) is available for patients with COVID-19, but is not included in this review.  Together, their review of therapeutics targeting the RAAS system is inaccurate and inadequate.

5: Section 6 includes the author’s development of anti-viral drugs.  The data is preliminary and does not include biological effects of these molecules. The section is not mature for publication.

The writing is wordy.  The manuscript could be 30% shorter without removing concepts.

Author Response

1: The review is not novel.  A search of PubMed for “COVID-19 AND Therapeutics AND Review” returns 1827 papers published 2020-2023. 

Thank you to the Editor comment on this point.

2: There are glaring omissions on the review, including but not limited to:-          - Long COVID.

- What was learned from failed versus successful drugs?  Without this, discussion of failed drugs is extraneous, and there’s no lessons for the next generation of COVID-19 drugs.

Response: The Conclusion and perspectives section is completely changed/modified to address this.

3: Hyper-inflammatory responses are the leading cause of morbidity and mortality from COVID-19, and so therapeutics targeting inflammation have been evaluated far beyond what this review includes. The new section on anti-inflammatory therapeutics to improve outcomes is inadequate (Section 3.1.6).  It does not relate therapeutic interventions to inflammatory processes in Section 2. It also ignores guidance related to NSAIDs or any other anti-inflammatory, provides essentially no molecular mechanism (all glaring omissions), and includes vaccines - which are not anti-inflammatories.  Even their new Table 3 includes anti-inflammatory therapeutics which are not discussed e.g. Actemra (Tocilizumab) and Kineret (Anakinra).

Response: The authors agree with the Editor comment: "I think that this is beyond the purview of this manuscript, which should focus on the virology."

4: Inaccurate information:

Section 3.1.5 inaccurately describes Captopril as an ACE2 inhibitor, when it is an ACE inhibitor. (corrected)

Meanwhile, the discussion on Ramipril, also an ACEi, states clinical studies for use of hypertensive medications in patients with COVID-19 are warranted. This is not accurate.  Assessment of hypertensive drugs has been done extensively, and guidance for use of ACEi and ARBs (hypertensive drugs that operate in the same biochemical pathway as ACE2 i.e. RAAS system) is available for patients with COVID-19, but is not included in this review.  Together, their review of therapeutics targeting the RAAS system is inaccurate and inadequate.

Response: corrected with more description and the following reference was added:

Mortality and Severity in COVID-19 Patients on ACEIs and ARBs-A Systematic Review, Meta-Analysis, and Meta-Regression Analysis. Front Med (Lausanne). 2022 Jan 10;8:703661. doi: 10.3389/fmed.2021.703661. PMID: 35083229; PMCID: PMC8784609.

5: Section 6 includes the author’s development of anti-viral drugs.  The data is preliminary and does not include biological effects of these molecules. The section is not mature for publication.

Response: This section was removed based on the Editor's request. 

Reviewer 2 Report (Previous Reviewer 1)

This is a well-balanced review, and the authors' revisions have successfully addressed my concerns. The revised manuscript is, in my opinion, suitable for publication. 

Author Response

Comments and Suggestions for Authors: This is a well-balanced review, and the authors' revisions have successfully addressed my concerns. The revised manuscript is, in my opinion, suitable for publication. 

Response: Thank you!

Round 2

Reviewer 1 Report (Previous Reviewer 3)

The authors have made substantial changes to the revised manuscript, and have addressed most of my concerns.  The manuscript is much more suitable for publication.  I have the following input for the author’s consideration.

(1) Line 799 – 801: In-silico screening demonstrated that captopril showed a good potential for inhibition of SARS-CoV-2 binding to ACE2 and the ADME analysis showed suitable pharmacokinetic properties [202].

Reference 202 does not support captopril preventing binding of SARS-CoV-2 to ACE2, and does not address ADME properties of this drug.  Either the reference or the statement is wrong.  Please correct this.

(2) The authors ignored one of my previous comments in their response. It is still my opinion the authors need to discuss the mechanisms of drugs (e.g. Actemra and Kineret) or molecular targets (e.g. CYP3A) that are in Table 3.  Currently the reader discovers drugs or drug targets in Table 3, but cannot relate them to SARS-CoV-2 because this discussion is absent.  Can you address this?

(3) Last line of the abstract: We introduce current efforts in designing novel nucleoside analogues for further investigation as potential viral inhibitors against the SARS-CoV-2 RNA-dependent RNA polymerase (RdRp).

The authors removed this topic from the manuscript, so this sentence should also be removed.

There is a grammar problem on lines 155-158: TMPRSS2 is expressed at the cell surface, and thus mediates S protein activation at the plasma membrane, whereas cathepsin L is present in the endosomal compartment and thus mediates activation occurs in the endolysosome.

There are spelling mistakes:

Line 18: ACE2, not ACE-2

Line 617: COVID, not COCID

Line 977: SARS-CoV-2, not SARSCoV-2

Author Response

(1) Line 799 – 801: In-silico screening demonstrated that captopril showed a good potential for inhibition of SARS-CoV-2 binding to ACE2 and the ADME analysis showed suitable pharmacokinetic properties [202].

Reference 202 does not support captopril preventing binding of SARS-CoV-2 to ACE2, and does not address ADME properties of this drug.  Either the reference or the statement is wrong.  Please correct this.

Response: The sentence is removed and is replaced by another one with the correct reference. 

(2) The authors ignored one of my previous comments in their response. It is still my opinion the authors need to discuss the mechanisms of drugs (e.g. Actemra and Kineret) or molecular targets (e.g. CYP3A) that are in Table 3.  Currently the reader discovers drugs or drug targets in Table 3, but cannot relate them to SARS-CoV-2 because this discussion is absent.  Can you address this?

Response: section 3.1.6., Anti-inflammatories, was changed to reflect the reviewer comment.  Actemra and Kineret with their mechanisms were discussed and the associated references were added (ref. 210 - 217).

(3) Last line of the abstract: We introduce current efforts in designing novel nucleoside analogues for further investigation as potential viral inhibitors against the SARS-CoV-2 RNA-dependent RNA polymerase (RdRp).

The authors removed this topic from the manuscript, so this sentence should also be removed.

Response: The sentence is removed from the abstract

There is a grammar problem on lines 155-158: TMPRSS2 is expressed at the cell surface, and thus mediates S protein activation at the plasma membrane, whereas cathepsin L is present in the endosomal compartment and thus mediates activation occurs in the endolysosome.

Response: The grammar is corrected

There are spelling mistakes:

Line 18: ACE2, not ACE-2  corrected

Line 617: COVID, not COCID  corrected

Line 977: SARS-CoV-2, not SARSCoV-2  corrected

This manuscript is a resubmission of an earlier submission. The following is a list of the peer review reports and author responses from that submission.

Round 1

Reviewer 1 Report

This review covers a lot of ground in the coronavirus field without going too deep, which is good for an overall view. The responses to SARS-CoV-2 by different kinds of human cells are discussed. The clinical trials for many antiviral drugs are summarized. As mentioned in the review, many drugs have been abandoned for the treatment of COVID-19. Thus, there is no point to write a lot of words about these drugs, maybe just make a table for them, and show their structures maybe. Overall, many sentences are redundant. Authors should make some effort to simplify the text.

Also, given the significant progress in mechanistic studies of RdRp and in computational methods, it is surprising that this review largely ignores developments in biochemistry that could pave the way for drug discovery.

1.      Line 13: Since COVID-19 is the disease caused by SARS-CoV-2, it is not suitable to write like this: “COVID-19 (SARS-CoV-2)”.

2.      Line 14: delete “within the host”.

3.      Lines 16 -19: Please separate this sentence. It’s a bit too long.

4.      Line 21: SARS-CoV-2 infection. Please change “in order to” to something else, like “to assist the identification of new drugs”.

5.      Line 32: belongs to.

6.      Line 34: what does ‘’ca.’’ mean?

7.      Table 1: ‘’collected’’ from respiratory tract of dead birds? For consistency, change “2019-nCoV” to “SARS-CoV-2”.

8.      Line 50: change “;” to “:”.

9.      Lines 54 – 56: the sentence “The human coronaviruses such as 229E…..(Table 1)” can be deleted. No need to repeat everything in table 1.

10.   Figure 1: please add the names of proteins, such as N, M, E….

11.   Line 69: please change “open reading frames, ORFs,” to “open reading frames (ORFs)”. And delete “required for RNA translation and viral machinery”.

12.   Line 76: change “Nonstructural proteins (NSPs)” to “Nonstructural proteins (nsps)”. Please change all nsp to lowercase.

13.   Line 79: COVID-19 is not a virus, it’s a disease. Also, this sentence in lines 78 – 80 is not necessary.

14.   Line 84: If the authors want to bring up the ORF1a and ORF1b, more information should be provided. At least illustrating how nsp12 was made by the frame-shifting mechanism.

15.   Table 2: 1) please indicate the full name of IFN. 2) the row of nsp6: formation of DMV. 3) Nsp7/8, cofactor of holo-RdRp. 4) nsp12, any evidence to support “primer dependent RdRp”? Authors should be careful about the primase activity of nsp8, many labs cannot see this activity. 5) nsp9, dimerization is not a function, it’s a property of nsp9, a mean to perform its function. Involvement in capping is likely an important function of nsp9.

16.   Line 102: delete “may”.

17.   Extra space in line 160.

18.   Line 220: SARS-CoV-2.

19.   Lines 293 – 294 and 306 – 307 are talking the same thing.

20.   Line 310: blood-brain barrier (BBB).

21.   Figure 2 is not cited anywhere. The ACE2 in the middle of the figure can be deleted. It’s confusing why it’s there.

22.   Line 329: SARS-CoV-2.

23.   Lines 352 – 356: the order of these two sentences should be inverted.

24.   Line 362: “therefore it is less toxic”. So hydroxychloroquine is less toxic than chloroquine because it’s more soluble?

25.   Figure 3: it will be better to indicate which drugs can effectively against MERS-CoV and SARS-CoV.

26.   Line 386: “replacing ATP with remdesivir and three additional nucleotides” cannot be understood by people who don’t know how remdesivir stalls RdRp.

27.   Lines 387 – 389: there is experimental data to support remdesivir is a better substrate (DOI: 10.1016/j.isci.2020.101849 )

28.   Line 403: confirm.

29.   Line 408: why show English and Greek at the same time?

30.   Lines 456 – 475: please re-write this sentence. Not easy to understand the meaning of it.

31.   Line 465: “eleven days for the lopinavir/ritonavir treatment” is confusing. If we looked at lines 435 – 437, lopinavir/ritonavir have no effects there, but they can clear the virus here.

32.   Line 471: HCoVs. And it’s better to just state the value of EC50 here.

33.   Line 479: HCoVs. Line 480, ca.? line 482: complex. Line 484/645: SARS-CoV-2. Line 495: DAAs.

34.   Line 497: delete “Interestingly”.

35.   Line 655: ACE-2 or ACE2? Line 660: delete the comma after “in silico”.

36.   Line 761: examine.

37.   Line 776: what are the scores of the other three compounds?

Author Response

Also, given the significant progress in mechanistic studies of RdRp and in computational methods, it is surprising that this review largely ignores developments in biochemistry that could pave the way for drug discovery.

response: a paragraph with six more references were added under the "5. Computer-Aided Drug Design" section. 

  1. Line 13: Since COVID-19 is the disease caused by SARS-CoV-2, it is not suitable to write like this: “COVID-19 (SARS-CoV-2)”. corrected
  2. Line 14: delete “within the host”. corrected
  3. Lines 16 -19: Please separate this sentence. It’s a bit too long. corrected
  4. Line 21: SARS-CoV-2 infection. Please change “in order to” to something else, like “to assist the identification of new drugs”. corrected
  5. Line 32: belongs to. corrected
  6. Line 34: what does ‘’ca.’’ mean? deleted
  7. Table 1: ‘’collected’’ from respiratory tract of dead birds? For consistency, change “2019-nCoV” to “SARS-CoV-2”. corrected
  8. Line 50: change “;” to “:”. corrected
  9. Lines 54 – 56: the sentence “The human coronaviruses such as 229E…..(Table 1)” can be deleted. No need to repeat everything in table 1. deleted
  10. Figure 1: please add the names of proteins, such as N, M, E…. corrected
  11. Line 69: please change “open reading frames, ORFs,” to “open reading frames (ORFs)”. And delete “required for RNA translation and viral machinery”. corrected
  12. Line 76: change “Nonstructural proteins (NSPs)” to “Nonstructural proteins (nsps)”. Please change all nsp to lowercase. corrected
  13. Line 79: COVID-19 is not a virus, it’s a disease. Also, this sentence in lines 78 – 80 is not necessary. corrected, and deleted
  14. Line 84: If the authors want to bring up the ORF1a and ORF1b, more information should be provided. At least illustrating how nsp12 was made by the frame-shifting mechanism. deleted
  15. Table 2: 1) please indicate the full name of IFN. 2) the row of nsp6: formation of DMV. 3) Nsp7/8, cofactor of holo-RdRp. 4) nsp12, any evidence to support “primer dependent RdRp”? Authors should be careful about the primase activity of nsp8, many labs cannot see this activity. 5) nsp9, dimerization is not a function, it’s a property of nsp9, a mean to perform its function. Involvement in capping is likely an important function of nsp9. corrected
  16. Line 102: delete “may”. deleted
  17. Extra space in line 160. deleted
  18. Line 220: SARS-CoV-2. corrected
  19. Lines 293 – 294 and 306 – 307 are talking the same thing. 306-307 deleted
  20. Line 310: blood-brain barrier (BBB). corrected
  21. Figure 2 is not cited anywhere. The ACE2 in the middle of the figure can be deleted. It’s confusing why it’s there. Figure 2 is cited and the sentence is deleted
  22. Line 329: SARS-CoV-2. corrected
  23. Lines 352 – 356: the order of these two sentences should be inverted. corrected
  24. Line 362: “therefore it is less toxic”. So hydroxychloroquine is less toxic than chloroquine because it’s more soluble? corrected
  25. Figure 3: it will be better to indicate which drugs can effectively against MERS-CoV and SARS-CoV. is indicated in the text
  26. Line 386: “replacing ATP with remdesivir and three additional nucleotides” cannot be understood by people who don’t know how remdesivir stalls RdRp. corrected and clarified
  27. Lines 387 – 389: there is experimental data to support remdesivir is a better substrate (DOI: 10.1016/j.isci.2020.101849 ) the reference was added
  28. Line 403: confirm. corrected
  29. Line 408: why show English and Greek at the same time? corrected
  30. Lines 456 – 475: please re-write this sentence. Not easy to understand the meaning of it. corrected
  31. Line 465: “eleven days for the lopinavir/ritonavir treatment” is confusing. If we looked at lines 435 – 437, lopinavir/ritonavir have no effects there, but they can clear the virus here. deleted and corrected
  32. Line 471: HCoVs. And it’s better to just state the value of EC50 here. corrected and the EC50 values are added
  33. Line 479: HCoVs. Line 480, ca.? line 482: complex. Line 484/645: SARS-CoV-2. Line 495: DAAs. corrected
  34. Line 497: delete “Interestingly”. deleted
  35. Line 655: ACE-2 or ACE2? Line 660: delete the comma after “in silico”. deleted
  36. Line 761: examine. corrected
  37. Line 776: what are the scores of the other three compounds? as the authors mentioned in the Review: The results of the proposed sulfamoyl benzoate derivatives revealed more steps to be taken for optimization of the structure to its target enzyme. The molecular docking scores suggested that only three among the six compounds tested have potential for improvement (Figure 7, compounds A-C), but still scored overall less negative in comparison to the reference known repurposed drugs. So, these section was just to present the proposed structures.

Reviewer 2 Report

In the manuscript titled “Insights into COVID-19: Perspectives in drug remedy and host cell responses”, after a short introduction regarding classification and structure of coronaviruses, the authors list a series of well known host cell responses. In the second part of this manuscript are reported a series of compounds as potential drug candidates and some of them already in clinical use. In the third part it is reported only one pathway of internalization of SARS-CoV-2 as potential target: the Clathrin-mediated endocytosis. The last part, concerning computer-aided drug design focalized on RNA dependent RNA Polymerase (RdRp), reports only one study of docking calculation on RdRp  carried out by the same authors.

The manuscript is only another among the high number of other reviews already published on this topic. In fact in PubMed are present at least 107 reviews, with the keywords used from the authors in this manuscript.  Novelty is absent. All the molecules reported in this manuscript were already taken into account in previously published reviews, often in a deeper manner. In figure 3 structure of Daclatasvir is wrong. It is the dimethylester and not the dicarboxylate dianion, furthermore the stereochemistry of  a stereogenic center is wrong (see structure in  PubChem:CID 25154714).

The part of “drug design computational approach” is limited to six uracil-sulfonamidic derivatives studied from the same authors, not as potential drugs on SARS-CoV-2, but as potential anticancer agents on human ribonucleotide reductase.

PubMed report many important computational study on SARS-CoV-2 targets, which are here not reported, as expected in a review aticle.

The last paragraph “Drug targeting of SARS-CoV-2 RdRp”, as observed for the previous one, is limited to introduce the results obtained from the same authors, through docking calculation, of the uracil-sulphonamido derivatives on RdRp.

Although, reporting a single study in a paragraph of a review is useless, it presents many issues:

1) the RdRp used in simulations (7BTF) is the native enzyme without the presence of a known ligand in the active site or better the presence of RNA growing  chain in which is bound a known inhibitor, in order to have enough space to accept a potential inhibitor. Alternatively you need to perform a molecular dynamic simulation on the native enzyme with the potential ligand, in order to create a correct pocket in the active site.

2) The resolution of used native enzyme (7BTF) is close to 3 Å, which is barely accetable. It would have been better to use the structure 7BV2 which has a better resolution (2.5 Å) and  it is bound the template-primer RNA and phosphate form of  GS441524 (remdesivir metabolite) making the enzyme prone in the active site to guest an inhibitor.

3) Probably, as for remdesivir, molecules A-F need to be bio-activated by removing the ester-sulfonamido moiety and triphosphorylated by cell enzymes to be integrated in the growing RNA chain. Unfortunately, the remaining unit, after removing ester-sulfonamido moiety is uracil, which is the natural nucleoside and not inhibit the enzyme or acts as a mutagenizing agent that causes an “error catastrophe” during viral replication as in the case of active remdesivir metabolite (GS441524-triphosphate) or other analogues as molnupiravir in which, the isobutyryl-ester moiety is removed and the nucleoside analog β-D-N4-hydroxycytidine triphosphorylated is readly incorporate in RNA.

For the reasons reported above, this docking approach is vain.

The critical discussion is absent in all the parts of this manuscript. In addition, the title does not reflect exactly what is reported in the text

In conclusion, the manuscript is poor. The bibliographic research concerning paragraph 4 and subsequent, is practically absent, in fact the authors have cited and reported only one their study, which is not eligible for a review. The conclusions are “inconclusive”. This manuscript does not improve knowledge on the topic.

The manuscript cannot be accepted for publication.

Author Response

In figure 3 structure of Daclatasvir is wrong. It is the dimethylester and not the dicarboxylate dianion, furthermore the stereochemistry of  a stereogenic center is wrong (see structure in  PubChem:CID 25154714). the dimethylester was corrected, however, the authors confirmed the accuracy of the chiral centers. Please advise if you still consider a specific stereogenic center is wrong.

The part of “drug design computational approach” is limited to six uracil-sulfonamidic derivatives studied from the same authors, not as potential drugs on SARS-CoV-2, but as potential anticancer agents on human ribonucleotide reductase.PubMed report many important computational study on SARS-CoV-2 targets, which are here not reported, as expected in a review article.

under the section "5. Computer-Aided Drug Design", a paragraph was added with six references for reported important computational studies.

1) the RdRp used in simulations (7BTF) is the native enzyme without the presence of a known ligand in the active site or better the presence of RNA growing  chain in which is bound a known inhibitor, in order to have enough space to accept a potential inhibitor. Alternatively you need to perform a molecular dynamic simulation on the native enzyme with the potential ligand, in order to create a correct pocket in the active site.

this part of the Review is mainly to introduce these molecules and their structures, and to pointed out our current effort to develop potential inhibitors. We wrote: "The results of the proposed sulfamoylbenzoate derivatives revealed more steps to be taken for optimization of the structure to its target enzyme." 

2) The resolution of used native enzyme (7BTF) is close to 3 Å, which is barely accetable. It would have been better to use the structure 7BV2 which has a better resolution (2.5 Å) and  it is bound the template-primer RNA and phosphate form of  GS441524 (remdesivir metabolite) making the enzyme prone in the active site to guest an inhibitor.  thank you for the suggestion. We will consider this when developing this structures further.

3) Probably, as for remdesivir, molecules A-F need to be bio-activated by removing the ester-sulfonamido moiety and triphosphorylated by cell enzymes to be integrated in the growing RNA chain. Unfortunately, the remaining unit, after removing ester-sulfonamido moiety is uracil, which is the natural nucleoside and not inhibit the enzyme or acts as a mutagenizing agent that causes an “error catastrophe” during viral replication as in the case of active remdesivir metabolite (GS441524-triphosphate) or other analogues as molnupiravir in which, the isobutyryl-ester moiety is removed and the nucleoside analog β-D-N4-hydroxycytidine triphosphorylated is readly incorporate in RNA.

That is exactly what we mentioned at the end of the Review. On the last paragraph we wrote: "Further direction for this study is to modify the three best scoring compounds in order to enhance binding to the active site of the RdRp. An initial attempt of changing the position of the sulfamoylbenzoate moiety from the 5’ of the ribose ring to the 2’ position to allow for the phosphorylation of the 5’ hydroxy to the triphosphate form, the RdRp natural substrate form, had significantly improved binding (Mod Compounds in Figure 7). As the goal was to inhibit RdRp function, further studies including the mechanism of action, testing further pockets to identify best binding, and in vivo studies are priorities. Should these compounds be viable, this could introduce another avenue for SARS-CoV-2 patient treatment."

Reviewer 3 Report

The review “Insights into COVID-19: Perspectives in drug remedy and host cell responses” by Awad et al. addresses numerous advances in knowledge of SARS-CoV-2, including the taxonomy of this virus, its method of entry into host cells, human pathology, and most extensively, the status of novel or repurposed drugs to combat severity of COVID-19 symptoms.  Early advances in designing or improving existing anti-viral drugs using an in silico method is presented, with intriguing preliminary results.  I wish the authors good luck in advancing there in silico discoveries.

However, goals set out in the abstract are not satisfied – see major concerns #1-2.

There are contradictions within the paper that need reconciliation. Major sections of the manuscript do not have a clear objective.

Major concerns:

1: The abstract states: "It provides publicly available knowledge on the chemical nature of proposed therapeutics and their target biomolecules in order to identify potentially new drugs for the treatment of COVID-19 infection".  However, the mechanism of action (MOA) for several therapeutics in section 3.1.2 are not clearly stated. Further, numerous therapeutics were not successful in patients with COVID-19. The authors do not organize what lessons were learned to design new therapeutics from the numerous failed candidates they include in this review. Thus the statement in their abstract is not satisfied, and it’s unclear why so much of the manuscript is dedicated to drugs without efficacy for SARS-CoV-2.  

2: Large portions of sections 1 and 2 involve inflammatory processes arising from COVID-19. However, anti-inflammatory drugs to control the cytokine storm are not included, with the exception of corticosteroids.  Why aren’t anti-inflammatories for COVID-19 treatment included more extensively in this review?  

3: The mechanism of SARS-CoV-2 entry into host cells is oversimplified as viral spike protein binding to ACE2 on the surface of host cells, followed up endocytosis (line 106 and section 4.1).  This is problematic in several ways:

-        The legend of figure 4 states: The internalization process for SARS-CoV-2 into cells is still under clarification. Which contradicts the well characterized endocytic process they’ve stated line 106 and section 4.1

-        Why aren’t inhibitors of endocytosis described in Sections 3.1.1 and section 4.1 efficacious with SARS-CoV-2?  This also contradicts SARS-CoV-2 entry by endocytosis.

-        TMPRSS2 receptor and camostat mesylate are included in section 3.1.2.  They are active in modifying SARS-CoV-2 spike protein to prime its binding to ACE2.  The authors do not describe priming of spike protein, nor discuss TMPRSS2 receptor and camostat mesylate, presumably because their description of the spike-ACE2 interaction followed by endocytosis is inadequate or incomplete.

-        The description of cathepsin L (lines 644-646) includes the S glycoprotein (i.e. SARS-CoV-2 spike protein) trimer fusing with the host cell membrane. This is inconsistent with spike binding host cell ACE2 (not the host cell membrane in lines 644-646), and discussion of the SARS-CoV-2 spike protein trimer is overwise absent from their review.

-        Thus the spike-ACE2 interaction and entry of SARS-CoV-2 into host cells is more sophisticated than the authors acknowledge, which provides several therapeutic opportunities that the authors have ignored.  

4: Figure 4 is not adequate.  What are “Endocytic inhibitors” (red dots) doing at different locations in the figure, especially with the relevance of endocytosis questioned in Major point 3.  What are the blue and green blobs?

5: Line 142 states that SARS-CoV-2 doesn’t replicate in dendritic cells, but dendritic cells are included in Figure 2, and described as a cell type with clinical manifestations due to SARS-CoV-2 infection via ACE2 expression. Line 142 and Figure 2 contradict each other.  Also, RAAS is included in the legend of Figure 2, but is not in the Figure. Figure 2 needs to align with (a) the text (b) the Legend.

Author Response

1: The abstract states: "It provides publicly available knowledge on the chemical nature of proposed therapeutics and their target biomolecules in order to identify potentially new drugs for the treatment of COVID-19 infection".  However, the mechanism of action (MOA) for several therapeutics in section 3.1.2 are not clearly stated. Further, numerous therapeutics were not successful in patients with COVID-19. The authors do not organize what lessons were learned to design new therapeutics from the numerous failed candidates they include in this review. Thus the statement in their abstract is not satisfied, and it’s unclear why so much of the manuscript is dedicated to drugs without efficacy for SARS-CoV-2.  

response: In the referred section, the authors described attempts that have been made towards identify potentially new drugs for the treatment of COVID-19 infection. They included the history for each repurposed drug. The sentence has been modifies to "It provides publicly available knowledge on the chemical nature of proposed therapeutics and their target biomolecules to assist the identification of potentially new drugs for the treatment of SARS-CoV-2 infection.

2: Large portions of sections 1 and 2 involve inflammatory processes arising from COVID-19. However, anti-inflammatory drugs to control the cytokine storm are not included, with the exception of corticosteroids.  Why aren’t anti-inflammatories for COVID-19 treatment included more extensively in this review?  

response: the authors believe that the concern of using the NSAIDs affect the risk of infection by SARS-CoV-2. It was reported that NSAIDs may impair the immune response to SARS-CoV-2 and delay disease resolution. Reference: J Virol. 2021, 95:e00014-21. doi: 10.1128/JVI.00014-21, thus we consider including them is not suitable in this review that focus on possible treatment.

3: The mechanism of SARS-CoV-2 entry into host cells is oversimplified as viral spike protein binding to ACE2 on the surface of host cells, followed up endocytosis (line 106 and section 4.1).  This is problematic in several ways:

-        The legend of figure 4 states: The internalization process for SARS-CoV-2 into cells is still under clarification. Which contradicts the well characterized endocytic process they’ve stated line 106 and section 4.1

-        Why aren’t inhibitors of endocytosis described in Sections 3.1.1 and section 4.1 efficacious with SARS-CoV-2?  This also contradicts SARS-CoV-2 entry by endocytosis.

-        TMPRSS2 receptor and camostat mesylate are included in section 3.1.2.  They are active in modifying SARS-CoV-2 spike protein to prime its binding to ACE2.  The authors do not describe priming of spike protein, nor discuss TMPRSS2 receptor and camostat mesylate, presumably because their description of the spike-ACE2 interaction followed by endocytosis is inadequate or incomplete.

-        The description of cathepsin L (lines 644-646) includes the S glycoprotein (i.e. SARS-CoV-2 spike protein) trimer fusing with the host cell membrane. This is inconsistent with spike binding host cell ACE2 (not the host cell membrane in lines 644-646), and discussion of the SARS-CoV-2 spike protein trimer is overwise absent from their review.

-        Thus the spike-ACE2 interaction and entry of SARS-CoV-2 into host cells is more sophisticated than the authors acknowledge, which provides several therapeutic opportunities that the authors have ignored.  

response: The following detailed description with associated references were added to section 2.2 of the Review

"The viral spike S glycoprotein consists of two subunits, S1 and S2. The C-terminal receptor-binding domain of the S1 subunit acts on the recognition of the ACE2, while S2 subunit helps the fusion of viral membrane into host cell membrane. Following this binding, a transmembrane serine protease (TMPRSS2) or other proteases such as furin or cathepsin L catalyze the cleavage of the S-protein [29]. TMPRSS2 is expressed at the cell surface, and thus mediates S protein activation at the plasma membrane, whereas cathepsin L is present in the endosomal compartment and thus mediates activation occurs in the endolysosome. Cell entry by SARS-CoV-2, therefore, depends on the target-cell proteases, with TMPRSS2 and cathepsin L remain the two major proteases involved in S protein activation [30].
Additional mediators have been reported to facilitate virus entry in some cell types. These include non-tyrosine kinase; neuropilin (NRP1), kidney injury molecule-1 (KIM1), glucose-regulated protein 78 (GRP78), metabotropic glutamate receptor subtype 2 (mGluR2), the heat shock protein A5 (HSPA5 or GRP78), the transmembrane glycoprotein CD147, and the G protein-coupled receptor mGluR2 (GRM2) [29, 30]. Other molecules that may act as co-factors in the entry process include ADAM metallopeptidase domain 17 (ADAM17), which is involved the shedding of the ACE2 ectodomain, and heparan sulfate (HS) which mediates ACE2 viral entry [31]. Further strategies of infections incorporate specific binding sites such as O-linked or N-linked glycans on the outer membrane of SARS-CoV-2, and hence other host molecules such as sugars and sialic acids may act as potential virus receptors [32]."

4: Figure 4 is not adequate.  What are “Endocytic inhibitors” (red dots) doing at different locations in the figure, especially with the relevance of endocytosis questioned in Major point 3.  What are the blue and green blobs?

response: Figure 4 is removed from the Review.

5: Line 142 states that SARS-CoV-2 doesn’t replicate in dendritic cells, but dendritic cells are included in Figure 2, and described as a cell type with clinical manifestations due to SARS-CoV-2 infection via ACE2 expression. Line 142 and Figure 2 contradict each other.  Also, RAAS is included in the legend of Figure 2, but is not in the Figure. Figure 2 needs to align with (a) the text (b) the Legend.

response: Neither TLR4‐expressing dendritic cells nor cell lines become activated by SARS‐CoV‐2. Notably, ectopic expression of ACE2 on dendritic cells leads to infection by SARS‐CoV‐2 and immune activation. Thus, SARS‐CoV‐2 triggers intracellular but not extracellular pattern recognition receptors, suggesting that immune responses are initiated upon direct infection or due to bystander inflammatory processes. Reference: Eur J Immunol 2022, 52:646-655. doi: 10.1002/eji.202149656

RAAS is removed from the legend of Figure 2 as we didn't intend to discuss it.

Round 2

Reviewer 3 Report

Comment 1: The authors did not address my original concern #1: (a) it’s unclear why so much of the manuscript is dedicated to drugs without efficacy for SARS-CoV-2. (b) The authors do not organize what lessons were learned to design new therapeutics from the numerous failed candidates they include in this review.

My point is if therapeutics are included in their paper (sections 3 and 4), the authors need to discuss why they were important to advance this field or why they’ve highlighted these studies, not just list them. This is most profound when therapeutics were not yet attempted or were not efficacious SARS-CoV-2 treatments, and are thus irrelevant or at least unjustified.

Inadequate discussion of therapeutics they’ve listed is a major shortcoming in the manuscript, and the authors did not address this in their revision.

Comment 2: Regarding my original comment #2: The authors confirm my original concern that large portions of sections 2 describe inflammatory processes arising from COVID-19, however, anti-inflammatory drugs to control the cytokine storm are not included.

The authors respond that NSAIDs affect the risk of infection by SARS-CoV-2 by impairing the immune response to delay resolution of SARS-CoV-2 infection, so they consider including NSAIDs not suitable in this review that focuses on possible treatments.

This is exactly my point.  The review is to describe possible treatments for SARS-CoV-2, however they consider inflammatory processes to be unsuitable for therapeutic intervention.  So, their lengthy discussion of SARS-CoV-2 associated inflammation is extraneous and off-topic. The revised manuscript does not address this shortcoming.

Author Response

Comment 1: The authors did not address my original concern #1: (a) it’s unclear why so much of the manuscript is dedicated to drugs without efficacy for SARS-CoV-2. (b) The authors do not organize what lessons were learned to design new therapeutics from the numerous failed candidates they include in this review.

response: extensive revisions based on this comment and new data and updated references have been added to section 3. Table 3 was also added.

Comment 2: Regarding my original comment #2: The authors confirm my original concern that large portions of sections 2 describe inflammatory processes arising from COVID-19, however, anti-inflammatory drugs to control the cytokine storm are not included.

response: 3.1.6. Anti-inflammatories was added to the manuscript.